# Role of Geometric Shape in Chiral Optics

**Philipp Gutsche [1,2,*]**, **Xavier Garcia-Santiago [3]**, **Philipp-Immanuel Schneider [4]**,
**Kevin M. McPeak [5]**, **Manuel Nieto-Vesperinas [6]** and **Sven Burger [2,4]**

1. Freie Universität Berlin, Mathematics Institute, 14195 Berlin, Germany
2. Zuse Institute Berlin, Computational Nano Optics, 14195 Berlin, Germany; burger@zib.de
3. Karlsruhe Institute of Technology, Institut für Theoretische Festkörperphysik, 76131 Karlsruhe, Germany; xavier.garcia-santiago@kit.edu
4. JCMwave GmbH, 14050 Berlin, Germany; philipp.schneider@jcmwave.com
5. Department of Chemical Engineering, Lousiana State University, Baton Rouge, LA 70803, USA; kmcpeak@lsu.edu
6. Instituto de Ciencia de Materiales de Madrid, Consejo Superior de Investigaciones Científicas, 28049 Madrid, Spain; mnieto@icmm.csic.es
* Correspondence: gutsche@zib.de

**Abstract:** The distinction of chiral and mirror symmetric objects is straightforward from a geometrical point of view. Since the biological as well as the optical activity of molecules strongly depend on their handedness, chirality has recently attracted high interest in the field of nano-optics. Various aspects of associated phenomena including the influences of internal and external degrees of freedom on the optical response have been discussed. Here, we propose a constructive method to evaluate the possibility of observing any chiral response from an optical scatterer. Based on solely the *T*-matrix of one enantiomer, planes of minimal chiral response are located and compared to geometric mirror planes. This provides insights into the relation of geometric and optical properties and enables identifying the potential of chiral scatterers for nano-optical experiments.

**Keywords:** optical chirality; mirror symmetry; helicity; optical scatterer

## 1. Introduction

It is usually a simple task to tell by eye whether an object is chiral or not: Achiral objects are superimposable onto their mirror image and, accordingly, they possess a mirror plane [1]. Recently, chiral scatterers have gained significant interest in nano-optics due to their potential to enhance the weak optical signal of chiral molecules [2–4]. Especially, the quantities of optical chirality and optical helicity as well as their relation to duality symmetry are subjects of current research [5]. The most established experimental technique in this field is the analysis of the circular dichroism (CD) spectrum which equals the differential energy extinction due to the illumination by right- and left-handed circularly polarized light [6].

In order to observe such chiral electromagnetic response, it seems to be obvious that geometrically chiral scatterers are required. However, it has been shown that extrinsic chirality, that is, a chiral configuration of the illumination and geometric parameters, yields comparable effects as intrinsically chiral objects [7]. By tuning the far-field polarization of the illumination, large chiral near-fields may even be generated in the viscinity of achiral objects [8]. In CD measurements, randomly oriented molecules are investigated, which can be classified by their *T*-matrix [9]. The latter has been used for quantifying the electromagnetic (e.m.) chirality, based on a novel definition of it [10].

However, the quantification of the geometric chirality is an elusive task [11] and even the unambiguous association of the terms right- and left-handed enantiomer of a general object is

impossible [12]. Different coefficients attempting to rate the chirality of an object are based on the maximal overlap of two mirror images [13] as well as the Hausdorff distance [14]. The choice of a specific coefficient determines the most chiral object [15], that is, there is no natural choice for quantifying geometric chirality. This also holds for the various figures of merit estimating the e.m. chirality. Similar correlations between geometric and optical properties are investigated with respect to the non-sphericity of arbitrary scatterers [16].

In this study, we start by transferring the simple procedure of finding a mirror plane to optics. Such symmetries are present in different mathematical descriptions as for example, block structures in the Mueller scattering matrix [17]. Here, we analyse the *T*-matrix and its associated geometric mirror symmetries by employing translation and rotation theorems of vector spherial harmonics. We illustrate this concept with numerical simulations of an experimentally realized gold helix. Different quantifications of the e.m. chirality are compared. Furthermore, the symmetry planes found in the optical response by our method are correlated to those of geometric origin. It is shown that the complex optical response, including higher order multipoles, yields mirror planes in the *T*-matrix which are not directly related to geometric symmetries.

In the following, we would like to briefly introduce the theory behind the methods described in this study. Further information may be found in Supplementary Materials.

The most general description of an isolated optical scatterer is the well-known *T*-matrix [18]. It relates an arbitrary incident field with the scattered field caused by the scattering object. The optical response to *any* incident field is included in the *T*-matrix. Accordingly, the following analysis of *T* is independent of *specific* illumination parameters such as the direction, polarization and beam shape. The goal of this study is to obtain insights into illumination-*independent* symmetries of the scatterer.

Usually, both the incident as well as the scattered field are given in the basis of vector spherical harmonics for computations with the *T*-matrix [19] (see Supplementary Materials). Physically observable quantities such as the scattered energy, the absorption, as well as the flux of optical chirality are readily computed from *T* [9]. In numerical simulations, *T* may be computed with high accuracy [20]. Knowing the response of the left-handed object $T_l$ enables the analytic computation of the response of its mirror image $T_r$:

$$T_r = \mathcal{M}_{xy}^{-1} T_l \mathcal{M}_{xy},$$ (1)

where we choose mirroring in the *xy*-plane $\mathcal{M}_{xy}$ without loss of generality (see Supplementary Materials for further details on notation). Note that the terminology of right- $T_r$ and left-handed $T_l$ is ambiguous, as pointed out before, and may be interchanged.

Since we aim to investigate arbitrary mirror planes, we note that an arbitrary plane is given by the three spherical coordinates of its normal: the inclination $\Theta$ and the azimuthal angle $\Phi$, as well as the distance $d$ from the origin. We define the according transformation $R(\Theta, \Phi, d)$ acting on the object as

$$R(\Theta, \Phi, d) = \mathcal{T}(\Theta, \Phi, d)\mathcal{R}_z(\Phi)\mathcal{R}_y(\Theta),$$ (2)

where $\mathcal{T}(\Theta, \Phi, d)$ is the translation of the *T*-matrix in the direction given by the angles and the distance and $\mathcal{R}_z(\Phi)$ and $\mathcal{R}_y(\Theta)$ are the rotations around the *z*- and *y*-axis, respectively [21] (see Supplementary Materials).

For a geometrically achiral object (see Figure 1a) there exists at least one transformation $R(\Theta, \Phi, d)$ such that $T_l = R(\Theta, \Phi, d)T_r R^{-1}(\Theta, \Phi, d)$. On the other hand, the lack of a geometric mirror plane of a chiral object Figure 1b implies that there exists no such transformation and that $T_l$ and $T_r$ do not coincide for any set of transformation parameters $(\Theta, \Phi, d)$. Note that this does not generally hold in the long wavelength limit, that is, the incident wavelength being much larger than the dimension of the scatterer, due to chiral dispersion [22].

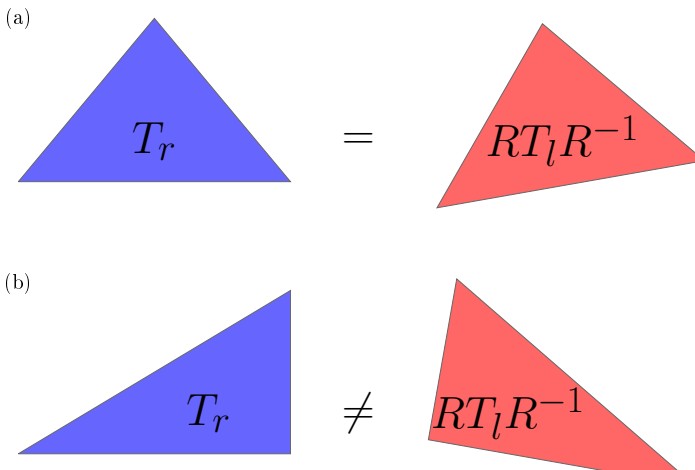

**Figure 1.** (**a**) The mirror image of an achiral object overlaps with its original after proper translations and rotations. This implies that the original *T*-matrix $T_r$ coincides with $T_l$ of the mirror object after the corresponding transformations *R*. (**b**) A chiral object and its mirror image are not congruent. If the object is much smaller than the incident wavelength, it usually exists a transformation *R* after which $T_l$ and $RT_rR^{-1}$ are equal. Note that the achiral isosceles triangle in (**a**) possess a mirror plane in 2D and that the asymmetric triangle in (**b**) is chiral only in 2D.

## 2. Results

For investigating the role of the geometric shape in nano-optics, it is of interest to identify those planes of highest symmetry of a chiral object: Although there is no mirror plane in a chiral object, a transformation may be identified in which the right- and left-handed *T*-matrices are closest to one another. Rating the closeness is done here by calculating the 2-norm of the difference of these two matrices. Accordingly, we introduce the coefficient $\chi_{\mathrm{TT}}$ which minimizes the difference between the *T*-matrices of mirror images as

$$\chi_{\mathrm{TT}} = \min_{(\Theta,\Phi,d)} \left|\left| T_l - R^{-1}(\Theta,\Phi,d)T_rR(\Theta,\Phi,d) \right|\right|_2 . \tag{3}$$

This means that for the mirror plane corresponding to minimal parameters $(\Theta_{\min}, \Phi_{\min}, d_{\min})$ of (3), the optical responses of the two mirror images are as similar as possible. In other words, the mirror images are hardly distinguishable. For an achiral object $\chi_{\mathrm{TT}}$ vanishes since there exists a transformation for which the mirror images are identical.

Obviously, the choice of the norm is not unique and other quantifications of similarity of the mirror images could be defined (see Supplementary Materials for the physical relevance of the 2-norm). A recently introduced coefficient $\chi_{\mathrm{SV}}$ is, for example, based on the singular-value decomposition of the *T*-matrix in the helicity basis [10]. Alternatively, the angular-averaged differential energy extinction $\chi_{\mathrm{CD}}$ due to illuminating with either right- or left-handed circularly polarized plane waves is experimentally accessible as the CD spectrum.

In order to exemplary introduce our formalism and compare it to previous work, we investigate a nano-optical device numerically. The finite element method is employed to accurately simulate the electromagnetic properties due to incident monochromatic light. Within this study we use the commercial FEM package JCMsuite [23]. In postprocessing, the *T*-matrix is computed by decomposing the scattered field into vector spherical wave functions [20] from illumination with 150 plane waves with randomly chosen parameters (see Supplementary Materials).

In Figure 2, we compare simulations of the aforementioned three coefficients quantifying the e.m. chirality for a gold helix as realized experimentally [24]. The helix is constructed on the surface of a cylinder with height 230nm and radius 60nm (see Supplementary Materials). The CD spectrum $\chi_{\mathrm{CD}}$ shows zero values at incident wavelengths of $\lambda = 615\,$nm and $\lambda = 1070\,$nm. If only these wavelengths

were analyzed, one could draw the conclusion that an achiral object is investigated. This contradicts the goal of this study to obtain insights into illumination-*independent* symmetries of the scatterer—for illuminations with $\lambda = 615$ nm and $\lambda = 1070$ nm, the scatterer seems to be geometrically achiral which is obviously not the case. Nevertheless, CD makes the chiral geometric nature of the helix visible as a maximum at 823 nm and a minimum at 1452 nm of smaller amplitude. For a helix with an opposite twist—that is, the mirror image—the roles of the extrema are interchanged.

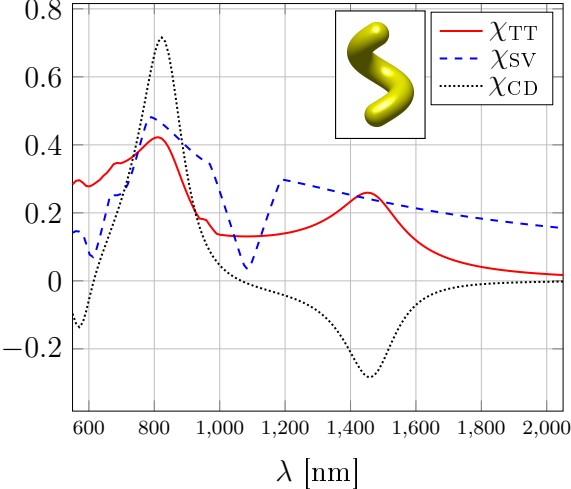

**Figure 2.** Chiral response of a gold nano-helix depending on the incident wavelength $\lambda$. The angular averaged differential extinction of circularly polarized plane waves $\chi_{\mathrm{CD}}$ (black dotted line) vanishes at 615 nm and 1070 nm which could be interpreted as achirality of the studied object. The electromagnetic chirality coefficient $\chi_{\mathrm{SV}}$ (dashed blue line) is based on the singular values of the $T$-matrix in the helicity basis. Values below 0.1 at 610 nm and 1085 nm indicate nearly achiral optical response. However, the minimal difference $\chi_{\mathrm{TT}}$ (red solid line) between $T_r$ and $RT_l R^{-1}$ reveals that the helix is chiral at all wavelengths. Its maxima correspond to those of $\chi_{\mathrm{CD}}$ and are, hence, observable.

On the other hand, the coefficient $\chi_{\mathrm{SV}}$ is normalized by the average interaction strength of the $T$-matrix at each wavelength. This yields a fairly flat spectrum with two narrow minima below 0.1 at the two $\lambda$ for which $\chi_{\mathrm{CD}} = 0$. These minima are not present in the minimized $\chi_{\mathrm{TT}}$ introduced in (3). However, the maxima of this latter coefficient are in accordance with the experimentally observable CD extrema ($\chi_{\mathrm{CD}}$). In the long wavelength regime, all three coefficients tend to zero as expected for point-like particles due to vanishing off-diagonal elements in the $T$-matrix.

The minimization in the three-dimensional parameter space in (3) is carried out using Bayesian optimization [25] (see Supplementary Materials). Since the shape of the minimized function highly depends on the actual object, the Bayesian approach is well suited for finding a global minimum. The parameters $(\Theta_{\min}, \Phi_{\min}, d_{\min})$ of the optimized value are related to geometric mirror planes. In Figure 3a, the planes following from the respective transformation $R(\Theta_{\min}, \Phi_{\min}, d_{\min})$ of the $xy$-plane are plotted for all incident wavelengths from 550 nm to 2.05 μm. The inclination $\Theta$ and azimuthal angle $\Phi$ are given in the shown coordinate system which is centered at the centroid of the helix.

We identify three distinct classes shown in blue, red and green. These correspond to planes which are parallel and perpendicular to the helix axis, as well as tilted by a small angle $\Theta$ from the horizontal position, respectively. The dark grey plane corresponds to the minimal geometric parameters which will be explained in the following paragraphs. Details on the optimization such as challenging flat behaviour for translations from the centroid and, on the obtained minimizing parameters, are given in Supplementary Materials. Note that the minimization required to obtain the illumination-independent coefficient $\chi_{\mathrm{TT}}$ involves significantly higher numerical effort than the simple averaging for $\chi_{\mathrm{CD}}$ for which most information contained in $T$ is ignored.

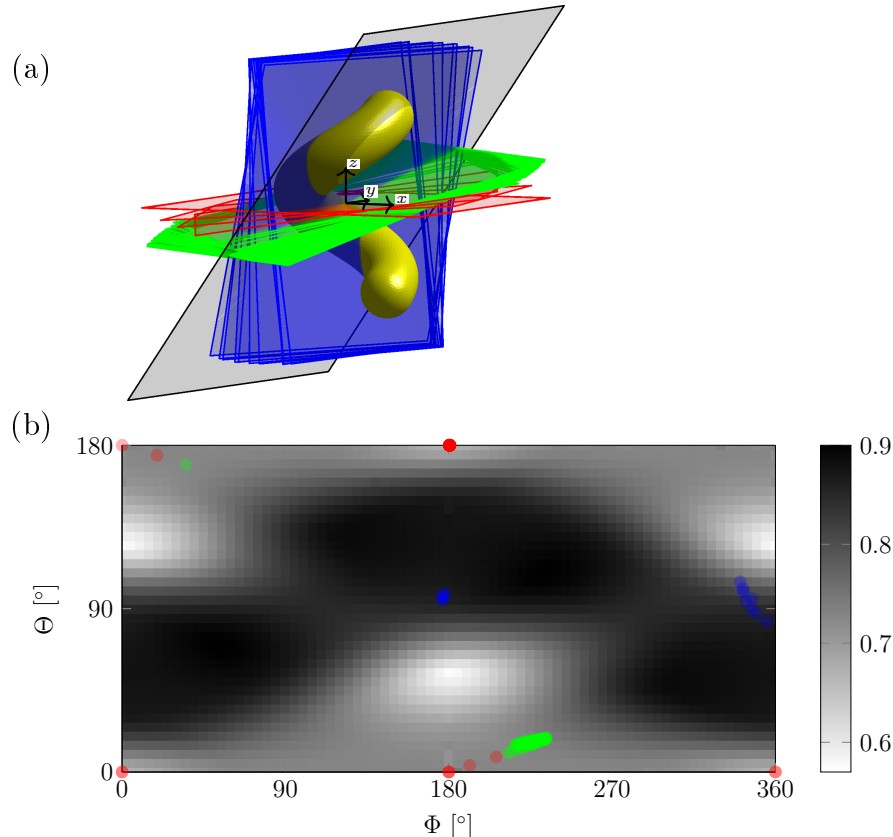

**Figure 3.** (**a**) Transformed $xy$-planes (blue, red, green) corresponding to minimal $\chi_{\mathrm{TT}}$ computed from $T$-matrix of the gold helix (yellow). Planes for all incident wavelenghts $\lambda \in [550, 2050]$ nm are shown. The dark grey plane corresponds to minimal $\chi_{\mathrm{GE}}$. (**b**) Geometric chiral coefficient $\chi_{\mathrm{GE}}(\Theta, \Phi)$ for the helix and its mirror image which is rotated around the centroid (grey colormap). The minimal value of 0.57 belongs to the dark grey plane in Figure 3a. Angles of the colored planes are shown by circles.

Next, we compare the findings on the symmetry based on the optical $T$-matrix to those stemming from purely geometric properties. As discussed previously, there is no coefficient which unambiguously rates the geometric chirality of an object. We choose a coefficient $\chi_{\mathrm{GE}}$ based on the overlap of the left- $O_l$ and right-handed $O_r(\Theta, \Phi, d)$ object, where the latter results from mirroring $O_l$ at the $xy$-plane and transformation with $(\Theta, \Phi, d)$. Namely, the volume $V$ of the overlap is compared to the volume of the object [13] (see Supplementary Materials):

$$\chi_{\mathrm{GE}}(\Theta, \Phi, d) = 1 - \frac{V\left(O_l \cap O_r(\Theta, \Phi, d)\right)}{V(O_l)}. \tag{4}$$

This coefficient vanishes for achiral objects as required for a degree of chirality [14].

Figure 3b displays the geometric chirality coefficient $\chi_{\mathrm{GE}}(\Theta, \Phi, 0)$ for planes rotated around the centroid of the helix as a grey colourmap. Dark regions with large values of $\chi_{\mathrm{GE}}$ indicate a vanishing overlap between the two mirrored helices. Note that for large distances to the origin $d \to \infty$, the mirror images do not overlap and $\chi_{\mathrm{GE}} = 1$. However, this is always possible no matter if the object is chiral or not. As in the case of $\chi_{\mathrm{TT}}$, the parameter points of interest of $\chi_{\mathrm{GE}}(\Theta, \Phi, d)$ are those corresponding to a minimum: The minimum 0.57 in Figure 3b occurs at $(180°, 55°)$ and $(0°, 125°)$ which show the intrinsic chiral property of the investigated helix. These two minima are equivalent since a finite helix is $C_2$ symmetric. The corresponding transformed $xy$-plane is shown in dark grey in Figure 3a.

Alongside the geometric coefficient $\chi_{\mathrm{GE}}$, the planes identified for the minimized $T$-matrix difference are shown as colored circles in Figure 3b. The colors (red, blue and green) of these circles are

the same colors used for the planes, that is, a direct comparison of the angle parameters is possible. As seen, the planes are ranked according to their $\Theta$ values: The perpendicular class 1 (blue) has $\Theta \in [83, 105]°$ The flat planes belonging to class 2 (red) show $\Theta \in [0, 8.5]°$ and $\Theta \in [174, 180]°$ and the tilted class 3 (green) has $\Theta \in [10, 19]°$ and $\Theta = 170°$.

## 3. Discussion

None of the three optical symmetry planes is directly related to the geometric mirror plane of the helix. However, Figure 3b enables the comparison of geometric and optical symmetries. In order to further analyze the optical response, we show the wavelength-dependent classification of the symmetry planes on top of Figure 4. The three classes correspond to sharply separated wavelength ranges: Class 1 is valid for $\lambda \in [550, 680]$ nm. For larger wavelengths $\lambda \in [680, 1025]$ nm, the *T*-matrix possesses the symmetry according to planes of class 2. Finally, in the long wavelength regime ($\lambda \in [1025, 2050]$ nm), the symmetry is in class 3.

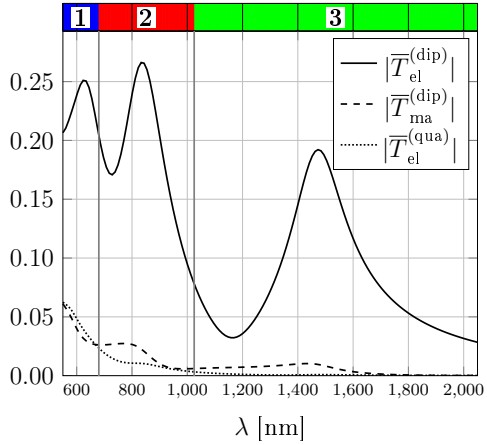

**Figure 4.** Wavelength-dependent classification of symmetry planes of *T*-matrix (top). Absolute value of averaged diagonal *T*-matrix entries corresponding to induced electric dipoles (solid), magnetic dipoles (dashed) and electric quadrupoles (dotted). The classes 3 (green), 2 (red) and 1 (blue) belong to decreasing wavelengths. Changes in symmetry of *T* are due to higher order multipoles.

The analysis in Figure 3b suggests that class 3 (green) is the closest one to the geometric mirror plane. This is further strengthened by the full angular spectrum of the optical chirality coefficient $\chi_{TT}$ (see Supplementary Materials). Accordingly, we find that the optical response is dominated by the geometric shape for long wavelengths. Obviously, the optics is dominated by the electric dipole moment in this regime which is also shown in Figure 4. Here, the mean of the diagonal entries of submatrices of the *T*-matrix are shown. These are proportional to the electric and magnetic dipole moments as well as to the electric quadrupole moments.

The three symmetry classes of the *T*-matrix occur close to three electric dipole peaks ($\lambda = 623$, 833, and 1473 nm) and are influenced by the anisotropy of the *T*-matrix. Truly chiral behaviour, as observed here, however, originates not from anisotropy but from coupling between electric and magnetic multipoles [26]. In Supplementary Materials, we elaborate on the complex interplay between these different contributions in the dipolar limit. Here, we limit the discussion to the main aspects of different multipolar contributions.

For large wavelengths with symmetry of class 3, the electric dipoles are much larger then any other induced multipole. In the intermediated regime of symmetry class 2, the magnetic dipole moment significantly increases. For short wavelengths with planes of class 1, the electric quadrupole moment is stronger than the magnetic dipole moment which yields the change in the optical symmetry. Higher order multipoles including mixed electric-magnetic moments are depicted in Supplementary Materials, in which it is shown that the dominant dipolar moments contribute additionally to the

variation of mirror planes. This elaborated study of multipolar resonances underlines again that the chiral response deviates from expectations due to a purely geometrical analysis of the scatterer.

## 4. Conclusions

In summary, we have introduced a method to obtain geometric mirror planes from the optical *T*-matrix of a scattering object. Accordingly, the optical effects of geometric structures such as metamaterials are analyzed [27]. We applied the procedure to an isolated gold helix and found correlations between the symmetry of its geometric shape and those of the optical response in the long wavelength regime. On the one hand, this confirms the expectation that instrinsic geometric chirality is directly related to an optically chiral response. On the other hand, for shorter wavelengths where higher multipoles are induced, mirror planes derived from the *T*-matrix do not coincide with the geometric mirror plane. This implies light-matter interactions whose symmetry cannot be explained simply by geometric chirality. Our method can be applied to all isolated scattering objects being chiral as the helix or achiral (see Supplementary Materials). It constructively identifies geometric planes of mirror symmetry in their optical response. This approach provides the basis for a detailed analysis of correlations between structural and spectral properties of nano-optical scatterers.

**Supplementary Materials:** The following are available online at http://www.mdpi.com/2073-8994/12/1/158/s1 and contain detailed information on *T*-matrix formalism, electromagnetic and geometric chirality coefficients, multipolar analysis, geometric model and optimization as well as the analysis of an achiral scatterer with our method.

**Author Contributions:** Conceptualization, P.G. and K.M.M.; methodology, P.G. and P.-I.S.; software, X.G.-S. and P.-I.S.; resources M.N.-V. and K.M.M.; writing—original draft preparation, P.G.; writing—review and editing, X.G.-S., P.-I.S., K.M.M., M.N.-V. and S.B.; supervision, M.N.-V. and S.B. All authors have read and agreed to the published version of the manuscript.

**Funding:** We acknowledge support from Freie Universität Berlin through the Dahlem Research School. This research was funded by the European Union's Horizon 2020 research and innovation programme under the Marie Sklodowska-Curie grant number 675745. This research was funded by the EMPIR programme co-financed by the Participating States and from the European Union's Horizon 2020 research and innovation programme under grant number 17FUN01 (BeCOMe). M. Nieto-Vesperinas acknowledges Spanish Ministerio de Ciencia, Innovación y Universidades, grants FIS2014-55563-REDC, FIS2015-69295-C3-1-P, and PGC2018-095777-B-C21.

**Acknowledgments:** We thank Martin Hammerschmidt for in-depth discussions on several topics.

**Conflicts of Interest:** The authors declare no conflict of interest.

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
