# Peer review of "Role of Geometric Shape in Chiral Optics"

_symmetry, doi:10.3390/sym12010158_

Round 1

Reviewer 1 Report

The manuscript studies the correlation between the geometrical definition of the object chirality and those based on its optical properties, described by a T-matrix. It is worth publishing in Symmetry, but a number of issues need to be addressed first.

The last sentence of Section 1. The authors should provide more justification, why the T-matrix is not chiral for very small sizes. The same applies to the caption of Figure 1. The first sentence of the conclusion should be softened or substantiated. In the studied example the T-matrix mirror planes are properly related to the geometric planes only after a detailed analysis, when the answer is known. I suspect that for an object with mirror plane the result would be less ambiguous (a minimum with zero value of the deviation), but even than a real example of such calculation is needed. The same affects the last sentence of conclusion. I could not find the description of the method that was used for simulation of the scattered field, which was then used (from 150 incident directions) to obtain T-matrix. The authors should describe it at least briefly. In Eq.(16) of the Supplementary and further, the authors think about (Tr – Tl) as a T-matrix for some “differential” scattering problem. Then Wsca and energy considerations are applied to it. The problem is that I am not sure that the difference of two T-matrices is always a T-matrix itself, i.e. it corresponds to any real scatterer. Thus, the obtained differential field has unclear physical meaning. So the authors should either remove this analogy or explain it in more details (with more justification), including the above caveats.

There are also several minor issues:

a) Here are a couple of papers that seems relevant to the effects of geometrical symmetries on that of optical properties. The authors may want to mention it in the Introduction:

- Kahnert FM, Stamnes JJ, Stamnes K. Application of the extended boundary condition method to homogeneous particles with point-group symmetries. Appl Opt 2001;40:3110–23.

- Yurkin MA. Symmetry relations for the Mueller scattering matrix integrated over the azimuthal angle. J Quant Spectrosc Radiat Transfer 2013;131:82–7.

b) Another recent paper also seems relevant, since its studies correlation between asymmetry and optical properties, but with respect to deviation from a sphere instead of chirality. It also builds both geometrical and optical measures of asymmetry. Interestingly, the geometrical measure is analogous to the ones used by the authors.

Romanov AV, Konokhova AI, Yastrebova ES, Gilev KV, Strokotov DI, Maltsev VP, et al. Sensitive detection and estimation of particle non-sphericity from the complex Fourier spectrum of its light-scattering profile. J Quant Spectrosc Radiat Transfer 2019;235:317–31.

c) When the Mxy first appears in Eq.(1), the authors should mention a bit more explicitly that the definition of this matrix is given in Supplementary.

d) The labels on the x-axis of Figure 2 are too close to each other. Please improve.

e) Eqs. (13),(14) of Supplementary. “2N(N+2)” should probably be changed to “N(N+2)+3”.

f) In Eq.(16) of the Supplementary (and further) T_l should probably be changed to R^(-1).T_l.R (as before that equation). Otherwise, it is completely not clear, why R disappears.

Reviewer 2 Report

This paper is short and succinct, introducing a rather abstract idea that involves introducing the concept of mirror planes to optical interactions. The paper is fine as a form of pure theoretical research but I do not expect any large degree of impact.  I am inclined to believe the results as they seem correct and match well-known paradigms of chiroptical interactions.

I find the second sentence of the manuscript confusing: ‘Recently chiral scatterers have gained high interest in nano-optics due to their potential to enhance the weak optical signal of chiral molecules’  What do the authors mean by this statement? It reads as though they mean that chiral scatterers in the vicinity of chiral molecules can enhance the optical activity signals of the chiral molecules (as in a two-body kind of interaction)? If this is what they mean then the references [2-4] do not fit with this statement. Can the authors clarify what they mean here and adjust the referencing accordingly.

Is the year correct (1940) for reference [6]?

Why have you discussed and mentioned the T-matrix numerous times, and then only formally introduced it on line 41? I would suggest that you introduce, define, and discuss the T-matrix as soon as you mention it. In fact, would the author’s not agree that Line 40 is a suitable place to end the Introduction section and Line 41-56 could form a ‘Theory’ or ‘Background’ section?

If the T-matrix calculations and results are independent of illumination parameters as you state, how does it account for the optical chirality associated with twisted laser beams possessing an optical orbital angular momentum that have recently been the topic of many studies by numerous researchers?   

Although adequate, I am sure the authors could produce a better looking set of figures for Figure 1.

Round 2

Reviewer 1 Report

The authors have addressed all my comments and improved the manuscript. There is only one typo that I have noticed: line 118: “corrsponding" -> “corresponding"